

# Differences in pollination success between local and foreign flower color phenotypes: a translocation experiment with *Gentiana lutea* (Gentianaceae)

Javier A. Guitián[1], Mar Sobral[2], Tania Veiga[1], María Losada[2], Pablo Guitián[1] and José M. Guitián[2]

[1] Departamento de Botánica/Facultade de Bioloxía, Universidade de Santiago de Compostela, Santiago de Compostela, A Coruña, Spain
[2] Departamento de Bioloxía Celular e Ecoloxía/Facultade de Bioloxía, Universidade de Santiago de Compostela, Santiago de Compostela, A Coruña, Spain

## ABSTRACT

**Background**. The adaptive maintenance of flower color variation is frequently attributed to pollinators partly because they preferentially visit certain flower phenotypes. We tested whether *Gentiana lutea*—which shows a flower color variation (from orange to yellow) in the Cantabrian Mountains range (north of Spain)—is locally adapted to the pollinator community.

**Methods**. We transplanted orange-flowering individuals to a population with yellow-flowering individuals and vice versa, in order to assess whether there is a pollination advantage in the local morph by comparing its visitation rate with the foreign morph.

**Results**. Our reciprocal transplant experiment did not show clear local morph advantage in overall visitation rate: local orange flowers received more visits than foreign yellow flowers in the orange population, while both local and foreign flowers received the same visits in the yellow population; thus, there is no evidence of local adaptation in *Gentiana lutea* to the pollinator assemblage. However, some floral visitor groups (such as *Bombus pratorum*, *B. soroensis ancaricus* and *B. lapidarius decipiens*) consistently preferred the local morph to the foreign morph whereas others (such as *Bombus terrestris*) consistently preferred the foreign morph.

**Discussion**. We concluded that there is no evidence of local adaptation to the pollinator community in each of the two *G. lutea* populations studied. The consequences for local adaptation to pollinator on *G. lutea* flower color would depend on the variation along the Cantabrian Mountains range in morph frequency and pollinator community composition.

Corresponding authors
Javier A. Guitián,
javier.guitian@usc.es
María Losada,
maria.cuquexo@gmail.com

## INTRODUCTION

Floral evolution is primarily driven by pollinators (*Bradshaw & Schemske, 2003*; *Whittall & Hodges, 2007*). The diversity in floral traits found in angiosperms, such as flower color, could be related with transitions in these traits (*Rausher, 2008*), as a result of different

selective pressures exerted by pollinators with strong preferences for certain characteristics (*Schemske & Bradshaw, 1999*; *Streisfeld & Kohn, 2007*) or by changes in pollinator community composition (*Ellis & Johnson, 2009*). Thus, floral diversification and speciation among closely related plant populations may result from the isolation and/or local adaptation to the most efficient pollinators, which increase the number of visits per plant depending on their flower preferences (*Harder & Johnson, 2009*; and references therein).

The adaptive maintenance of flower color variation is frequently attributed to pollinators, in part because they tend to promote assortative mating, i.e., preferentially visiting certain phenotypes (*Waser & Price, 1981*; *Stanton, 1987*; *Jones & Reithel, 2001*). However, many other reasons could explain the evolution of floral diversification, such as the potential uncoupling of the evolution of floral traits from pollinator-mediated selection (*Strauss & Whittall, 2006*). For example, pleiotropic effects on the biosynthetic pathways of floral pigmentation may drive the evolution to alternative flower colors (*Armbruster, 1993*; *Schemske & Bierzychudek, 2007*; *Cooley, Carvallo & Willis, 2008*).

Many studies support that animal pollinator preferences may cause selective pressures on flower color, which may fluctuate depending on the structure and/or composition of the pollinator community (*Melendez-Ackerman & Campbell, 1998*; *Gómez & Zamora, 2000*). In this context, reciprocal transplant experiments between populations can be a powerful tool to assess how changes in flower color preferences of the pollinator assemblage contribute to an adaptive advantage of the local morph over the foreign morphs (*Streisfeld & Kohn, 2007*).

*Gentiana lutea* L. shows flower color variation along the Cantabrian Mountains (Northern Spain). *Gentiana lutea* flowers are typically yellow (*G. lutea* var. *lutea*), though northwest Iberian populations have orange-flowering individuals, which constitutes a different variety of the species (*G. lutea* var. *aurantiaca; Renobales, 2012*). The pollinator assemblages are made up mostly of bumblebee species (*Veiga et al., 2015*; *Sobral et al., 2015*). Previous studies conclude that: (i) *Gentiana lutea* flower color variation does not result from adaptation to environmental factors, such as elevation, temperature, radiation, and precipitation (*Veiga et al., 2016*); (ii) *Gentiana lutea* is strongly dependent on pollinators for reproduction, and the plant has a positive relationship between the number of pollinator visits and the number of seeds it produces (*Losada et al., 2015*); (iii) there is a partial hybridization barrier among *G. lutea* color morphs (*Losada et al., 2015*); (iv) *Gentiana lutea* flower color influences pollinator visits and its variation is related to changes in pollinator community composition across populations, since part of this variation is explained by different pollinator selective pressures exerted on flower color among *G. lutea* populations (*Sobral et al., 2015*); and (v) the most abundant pollinators of our study species, *Bombus terrestris* and *B. pratorum* (*Veiga et al., 2015*), possess photoreceptors with low sensitivity to red colors; but other pollinators interacting with *G. lutea* plants, such as *Bombus lapidarius,* has high sensitivity to red colors despite the lack of photoreceptors required (*Peitsch et al., 1992*).

In order to determine whether *G. lutea* populations are locally adapted to the pollinator community, we evaluated the effect of local and foreign flower color morphs on pollinator visitation rate by means of reciprocal transplants in two *Gentiana* populations. More specifically, we tested whether there is an adaptive advantage in the local flower color

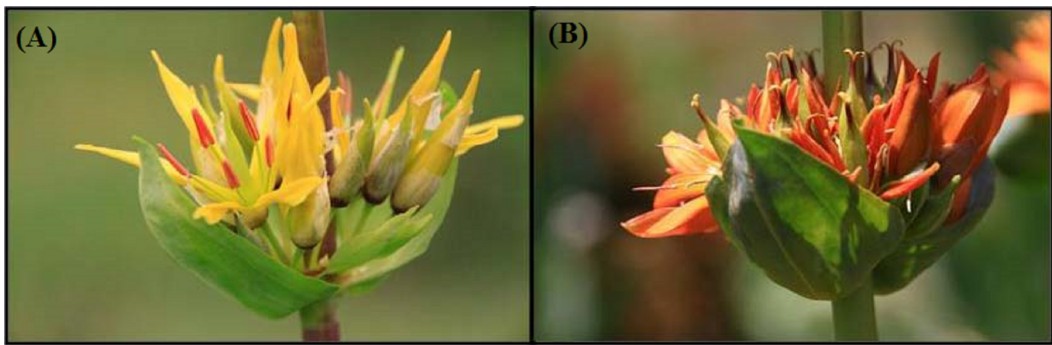

**Figure 1** **Color morphs of *Gentiana lutea* L. at Northern Spain.** (A) var. *lutea*, the yellow morph in Pontón location; and (B) var. *aurantiaca*, the orange morph in Ancares location.

morph relative to the foreign morph in pollinator visits by comparing the (i) overall pollinator assemblage, and (ii) different visitor groups separately. We predicted that the local flower color morph would receive more pollinator visits than the foreign morph, and that the two different color morphs would receive visits by different pollinator groups in their local populations.

## MATERIAL AND METHODS

### Plant species

*Gentiana lutea* L. is an herbaceous perennial geophyte (Fig. 1). During summer, the rhizome develops a flowering stalk of ca. 80 cm high, with a basal rosette formed by 4–8 oblong leaves. Flowers grow in two opposed cymose groups (each one with 15 flowers) at different stalk levels, above two elliptic leaves; and one flower crowns the top of the stalk. Flowers feature a rotate corolla with 5–7 lobules, 5–7 free stamens, and a fixed ovary with two nectaries on the base. The fruit is an ovoid capsule, holding elliptic and winged seeds of 3–4 mm (*Renobales, 2012*). This species needs pollinators—mainly bumblebees—to produce seed (see *Veiga et al., 2015*; *Sobral et al., 2015*).

*Gentiana lutea* flower color varies between orange and yellow (see Fig. 1) along the Cantabrian Mountains (Northern Spain; Fig. 2). The western populations (west of Somiedo Natural Park) have orange-flowering individuals, whereas the eastern populations (east of Puerto de Ventana) have yellow-flowering individuals; and the intermediate populations show individuals with flowers of a gradation between these two color morphs.

### Study site

The study was conducted in July 2012 at the northwest of the Iberian Peninsula in two populations, one located in Sierra de Ancares (orange-flowered population) and other in Puerto del Pontón (yellow-flowered population), separated by approximately 160 km (Fig. 2). Population size was estimated by the total number of *G. lutea* individuals (>3,000 in both populations).

Flower color of 75 haphazardly selected flowers (three flowers per plant, 25 plants) was measured the previous year in both populations by means of a spectrometer (USB2000+;

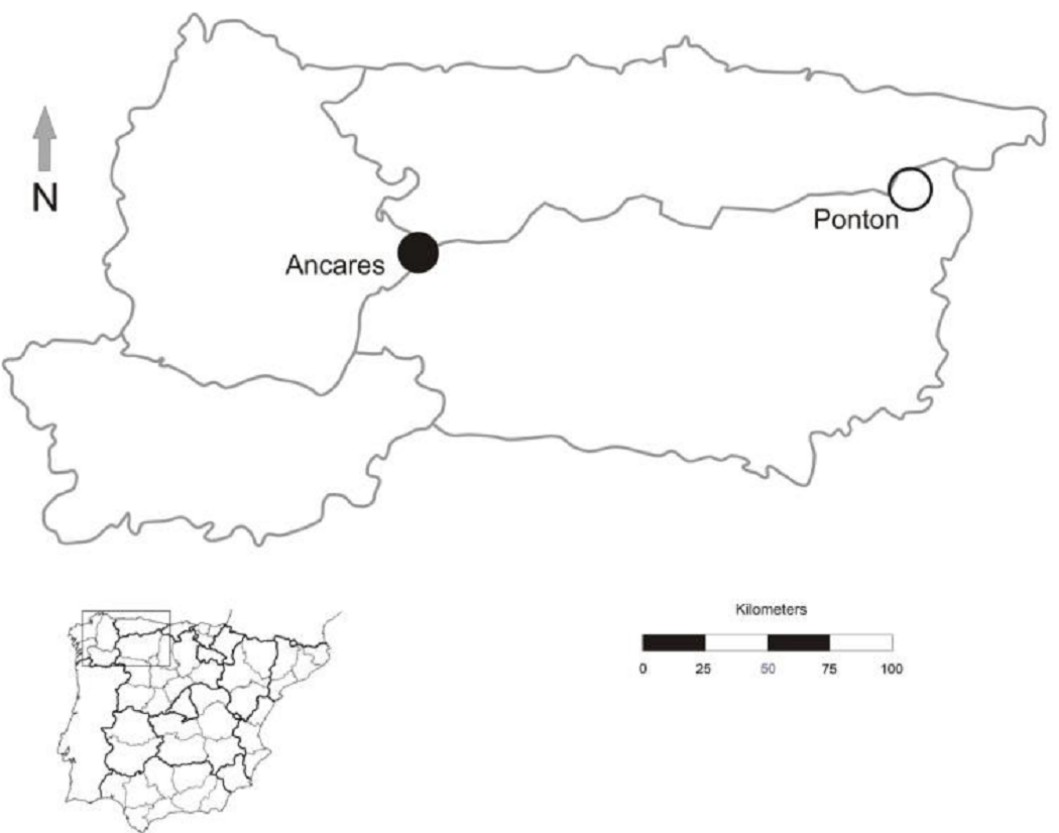

**Figure 2** **Geographic location of *Gentiana lutea* populations at Northern Spain in which reciprocal transplants were conducted.** Black dot, Ancares, the orange morph population; white dot, Pontón, the yellow morph population.

Ocean Optics, Inc., Dunedin, FL, USA) and petal color spectra were processed using the SpectraSuite® software (Ocean Optics, Inc., Dunedin, FL, USA). The CIELab Colorimetric System (*CIE P., 2004*) was chosen to describe flower color variation in the visible range of the electromagnetic spectrum, through three variables reduced by principal component analysis (for additional details, see *Veiga et al., 2015*): $L$ (brightness of color, from black to white), $a$ (red color variation, from green to red) and $b$ (yellow color variation, from blue to yellow). Individuals from the Sierra de Ancares population showed orange-colored flowers ($L = 16.42$; $a = 11.77$; $b = 23.61$), and from the Pontón population, yellow ones ($L = 19.64$; $a = 3.84$; $b = 27.84$).

Although certain pollinator species may detect UV light, a previous study found no differences among *G. lutea* populations and between individuals with different color morphs (orange or yellow, discernible by human eye) in the UV light range (*Veiga et al., 2015*). Thus, we presume that UV light does not drive local adaptation among populations, and consider flower color in both, the UV and visible light ranges in this study.

## Experimental design

In July 2012, a reciprocal transplant experiment was conducted to analyze local adaptation of color morphs to pollinator assemblage of both populations (thanks to the field

permission issued by the Environmental Territorial Service institution from León, Regional Government of Castilla and León, Territorial Delegation of Government of Spain—Identifier: 12_LE_325_RNA_PuebladeLilio_INV—Reference: 06.01.013.016/ROT/abp—File number: AEN/LE/103/12). Given the difficulty of transplanting plants in pots due to their extensive rhizomes, flowering stems used for the experiment were cut at the height of rhizome, placed in plastic containers with water (0.5 L), sealed with adhesive tape, and buried at ground level to prevent evaporation and ensure longer flower life. The effectiveness of this method was previously determined by checking the duration of flowering and pollination in a control population and verifying that the flowers were well-preserved in the first four days.

For the reciprocal transplant experiment, 15 flowering stems were cut in the orange population (Ancares) and transferred to the yellow population (Pontón) immediately after collection, while simultaneously, 15 yellow stems were cut in Pontón and transferred to the Ancares population (foreign transplant treatment). Additionally, 15 plants of each population (orange in Ancares and yellow in Pontón) were subjected to the same procedures (cut, transplanted, and buried) for the local transplant treatment.

The same day, plant containers were placed in each population within 1 m$^2$ squares, set in random layout by picking the grid point using a random number table. Additionally, 15 plants were haphazardly marked to serve as a control in both populations. Thus, each population contained 30 experimental plants and 15 control plants (45 plants per population, 90 plants overall). Note that the stem height, the number of floral whorls per inflorescence, and the number of flowers were measured in all plant individuals to correct for possible plant morphological effects in the experimental results.

## Pollinator censuses

We conducted 15 2-minute pollinator censuses on each of the 45 plants in both flower color morph populations over 3 days. According to previous studies, the observation sampling effort (30 min per plant, 1,350 min in each population) is appropriate to obtain an adequate representation of the pollinator spectra of this species and in this geographic area (see Materials and Methods section in *Veiga et al., 2015*; *Sobral et al., 2015*). For each census, the species that accessed the flowers and the number of flowers they visited were recorded.

Every day and before the census, the number of open flowers on each plant was counted to correct for the effect of floral display in pollinator attraction. Plant censuses were randomly conducted under sunny conditions between 0800 and 1800 h. The number of visits per plant (visitation rate per 30 min) was used as a measure of pollination success.

At the end of each census, a sample of pollinator taxa observed was collected and sent for further identification by entomological specialists from the University of Oviedo (Spain). Correcting any discrepancies between our visual identifications of bumblebee species and the lab determinations, species were grouped for analyses into eight morphological categories based on proboscis length (*Obeso, 1992*). The eight morphological groups were: group 1 (*Bombus terrestris, B. lucorum*); group 2 (*B. hortorum, B. jonellus*); group 3 (*B. pratorum, B. soroensis ancaricus, B. lapidarius decipiens*); group 4 (*Bombus wurflenii*);

group 5 (*B. mesomelas*); group 6 (*B. pascuorum*); group 7 (*B. (Psithyrus) rupestris*); group 8 (*B. hypnorum*).

## Statistical analyses

Several generalized linear models (GLM) were performed, where the number of visits per plant was the dependent variable. The locality (Ancares and Pontón), treatment (control, local transplant, and foreign transplant), pollinator group (described earlier), and the interactions among these factors were included as fixed effects. The models were performed for all pollinator species pooled together and for each of the eight pollinator groups independently. Residuals of the dependent variable were fitted in all models with a Poisson distribution and a logarithmic link function.

Additionally, pairswise contrasts were performed to analyze the differences on the visitation rate (number of visits per plant per 30 min) by the eight bumblebee groups between and within the transplant treatments and control group averaged for both destination localities. To test for local adaptation, the mean differences on visitation rate of group 1 and 3 bumblebees (the most abundant groups) were compared between each pair of treatments applied for the translocation experiment (foreign transplants, local transplants, controls), considering both localities (Ancares and Pontón). All analyses were performed using SPSS Statistics 20 (IBM Corp., Somers, NY).

## RESULTS

### Pollinator censuses

In total 951 insects (99% bumblebees belonging to the genus *Bombus,* and the remaining 1% belongs to the genus *Apis* or the family Vespidae) visited the plants (538 in Ancares, and 413 in Pontón), which represented 3,268 total cumulative flower visits (see Data S1). *Bombus terrestris* (group 1) and *Bombus pratorum* (group 3) were the main flower visitor of *Gentiana lutea* plants in Pontón, with 447 and 372 total cumulative visits respectively for each group (excluding control treatment, see Fig. 3). *Bombus soroensis ancaricus* + *Bombus lapidarius decipiens* (group 3) and *Bombus (Psithyrus) rupestris* (group 7) were the main flower visitors in Ancares (338 and 224 total cumulative visits respectively for each group, only for transplant treatments), being the latest group absent in Pontón, and *Bombus hypnorum* (group 8) was only present in this location (Fig. 3).

### Reciprocal transplants

In Ancares, a population of orange morphs, more visits per 30 min were observed to orange—than to yellow—flowering plants (32.5% versus 24.7% plants visited, respectively for each color morph); whereas in Pontón, a population consisting of yellow flowers, there was no difference in visitation rate between the two morphs (39.2% versus 41% plants visited, respectively for yellow and orange morphs; see Data S2). Number of visits per plant depended on the floral visitor group ($P < 0.001$; Table 1). Additionally, the number of visits per plant by floral visitor groups were affected differently per treatment: control, local transplant, and foreign transplant (as suggested by the significant treatment*floral visitors interaction; $P < 0.05$; Table 1), and also by locality (as suggested by the significant treatment*floral visitors*locality interaction; $P < 0.05$; Table 1).

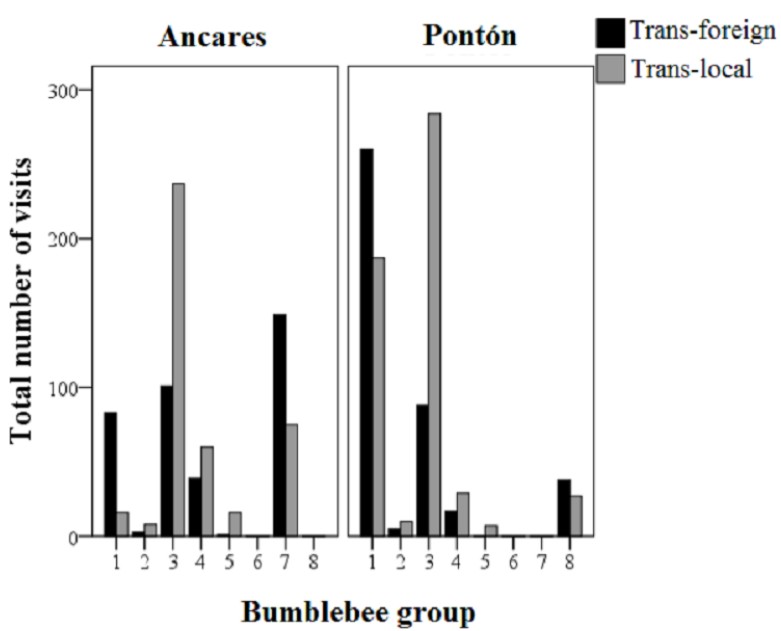

**Figure 3** **Total cumulative number of visits to the *Gentiana lutea* transplanted plants (black-colored bars, foreign; grey-colored bars, local) by each group of bumblebees in the two study sites at Northern Spain (Ancares with the orange morph and Pontón with the yellow morph).** Abbreviations: Trans-foreign, foreign transplant; Trans-local, local transplant. Floral visitor groups were the following: group 1, *Bombus terrestris*, *B. lucorum*; group 2, *B. hortorum*, *B. jonellus*; group 3, *B. pratorum*, *B. soroensis ancaricus*, *B. lapidarius decipiens*; group 4, *B. wurflenii*; group 5, *B. mesomelas*; group 6, *B. pascuorum*; group 7, *B. (Psithyrus) rupestris*; group 8, *B. hypnorum*.

Results of the GLM to analyze the locality and treatment effects on the number of visits per plant for each pollinator group showed different flower color preferences, but only in the groups 1 (*B. terrestris*) and 3 (*B. pratorum*, *B. soroensis ancaricus* + *B. lapidarius decipiens*; significant treatment factor; group 1: $P < 0.05$; group 3: $P < 0.01$; Table 2). Group 1 and group 3 pollinators visited control and local plants similarly in both color morph populations (Table 3 and Fig. 4).

Group 1 visited foreign transplant individuals more often than local transplant individuals, considering both locations (significant mean differences between foreign and local transplants; $P < 0.05$; Table 3). Group 1 pollinators visited control and local plants similarly in both color morph populations (Table 3 and Fig. 4A). However, the absolute number of visits by these pollinators was higher in Pontón (yellow morph population) than in Ancares (orange morph population).

Group 3 visited local transplants more often than foreign transplants in both locations (no significant treatment*locality interaction; Table 2). The number of visits by group 3 to control and foreign plants was equal considering both populations (Table 3 and Fig. 4B). Once again, the absolute number of visits by these pollinators was higher in Pontón (yellow-flowered population) than in Ancares (orange-flowered population).

**Table 1** **Results of the generalized linear model to analyze the effect of color morphs of *Gentiana lutea* on the visitation rate (number of visits per plant per 30 min) by eight bumblebee groups in two areas of Northern Spain (the orange color morph was present at the Ancares site, whereas the yellow morph was present at the Pontón site).** We marked in bold the statistically significant factors ($P < 0.05$). Factors: locality (Ancares and Pontón, orange and yellow morph population respectively); treatment (control, local transplants, and foreign transplants); and floral visitor groups (group 1, *Bombus terrestris*, *B. lucorum*; group 2, *B. hortorum*, *B. jonellus*; group 3, *B. pratorum*, *B. soroensis ancaricus*, *B. lapidarius decipiens*; group 4, *B. wurflenii*; group 5, *B. mesomelas*; group 6, *B. pascuorum*; group 7, *B. (Psithyrus) rupestris*; group 8, *B. hypnorum*).

| Factors | Wald Chi-Square | *d.f.* | *P* |
|---|---|---|---|
| Locality | 0.702 | 1 | 0.402 |
| Treatment | 0.110 | 2 | 0.946 |
| Treatment * Locality | 4.336 | 2 | 0.114 |
| **Floral visitors** | 66.548 | 3 | **<0.001** |
| **Treatment * Floral visitors** | 16.703 | 6 | **0.010** |
| **Treatment * Floral visitors * Locality** | 19.573 | 9 | **0.021** |

**Table 2** **Results of the generalized linear model to analyze the effect of locality, treatment, and the corresponding interaction between both, on the visitation rate (number of visits per plant per 30 min) by two bumblebee groups (group 1 and 3) in two areas of Northern Spain (Ancares and Pontón, the orange and yellow morph population respectively).** We marked in bold the statistically significant factors ($P < 0.05$). Factors: locality (Ancares and Pontón, orange and yellow morph population respectively); treatment (control, local transplants, and foreign transplants); and floral visitor groups (group 1, *Bombus terrestris*, *B. lucorum*; group 3, *B. pratorum*, *B. soroensis ancaricus*, *B. lapidarius decipiens*). Note that we only included results of the visitor groups that showed significant model effects (Group 1, Group 3).

| Floral visitors | Factors | Wald Chi-Square | *d.f.* | *P* |
|---|---|---|---|---|
| Group 1 | **Locality** | 32.222 | 1 | **<0.001** |
| | **Treatment** | 6.181 | 2 | **0.045** |
| | Treatment * Locality | 0.692 | 2 | 0.707 |
| Group 3 | Locality | 1.867 | 1 | 0.172 |
| | **Treatment** | 16.331 | 2 | **<0.001** |
| | Treatment * Locality | 0.114 | 2 | 0.945 |

## DISCUSSION

Our results show that *Gentiana lutea* plants from both locations face different sets of flower visitors with different color preferences—indicated by differences in visitation rate—suggesting that these differences may play a role in floral color divergence patterns. We found that the orange-flowering plants may be locally adapted to the pollinator assemblage since they received more visits than the yellow-flowering transplants in Ancares, the population of orange morphs. However, we found no evidence of local adaptation of yellow-flowering plants to the original population since the number of bumblebee visits was similar to the orange-flowering transplants in Pontón. This result differs from previous research, which suggested that flower color variation among *G. lutea* populations is related to selective pressures exerted by different pollinator assemblages (*Sobral et al., 2015*).

Several possible explanations may help to clarify our findings. The different responses obtained in the Ancares and Pontón populations are likely due to variation in the pollinator

**Table 3  Results of pairwise contrasts to analyze the mean differences on the visitation rate (number of visits per plant per 30 min) by bumblebee groups 1 and 3, averaged for two areas of Northern Spain (Ancares and Pontón, the orange and yellow morph population respectively) and per treatment (control, local transplant, foreign transplant).** We marked in bold the statistically significant mean differences between the pair of treatments ($P < 0.05$). Abbreviations: Trans-local, local transplants; Trans-foreign, foreign transplants; and control, unmanipulated groups. Note that we only included results from pairwise contrasts that showed significant mean differences between pair of treatments (i.e., pollinator groups 1 and 3).

| Floral visitors | Pair of treatments | | Mean-difference | S.E. | $d.f.$ | $P$ |
|---|---|---|---|---|---|---|
| Group 1 | Control | Trans-foreign | −5.090 | 2.801 | 1 | 0.069 |
| | Control | Trans-local | 1.390 | 2.331 | 1 | 0.551 |
| | **Trans-foreign** | **Trans-local** | 6.480 | 2.733 | 1 | **0.018** |
| Group 3 | **Control** | **Trans-foreign** | 18.720 | 4.369 | 1 | **<0.001** |
| | Control | Trans-local | 8.210 | 4.992 | 1 | 0.100 |
| | **Trans-foreign** | **Trans-local** | −10.510 | 3.736 | 1 | **0.005** |

assemblage and/or floral visitor preferences between the two areas (as experimentally demonstrated, *Jones & Reithel, 2001*). Ancares and Pontón have different pollinator assemblage and the species from the Ancares population may prefer local orange-flowering plants, while the species from Pontón population clearly show no preference for one of the two color morphs. Geographic variation in floral visitor preferences can result from changes in pollinator composition, abundance, and diversity (*Price et al., 2005*), even different plant populations that share the same pollinator group receive different proportions of flower visits from each species or functional group (*Fenster et al., 2004*; *Tastard et al., 2014*). Thus, pollinator-dependent foraging preferences may cause shifts in the optimal floral phenotypes (see *Gómez et al., 2009*; and references therein).

Within the pool of *G. lutea* flower visitors in the Ancares population, the most abundant species *(Bombus terrestris)* has low sensitivity for red color detection (*Peitsch et al., 1992*; *Briscoe & Chittka, 2001*), which probably explains the reduction in this species visitation rate when increasing orange-flowering individuals (Fig. 4A). We suspect that other pollinator species with probably higher sensitivity to red color may increase fitness of the orange-flowering plants. Therefore, the fact that they perceive red color does not imply that they necessarily would prefer this color over others. In fact, *B. lapidarius,* present in other orange *G. lutea* (JA Guitián, pers. obs., 2012), shows sensitivity for red colors; although it lacks red color receptors (*Kugler, 1943*; *Peitsch et al., 1992*). If pollinator species with a higher sensitivity or innate preferences for red colors are present, orange-flowered plants would increase their fitness through the rising visitation rate by these pollinators. Therefore, variation of the pollinator community composition might affect the selective pressures exerted on *G. lutea* flower color, and ultimately affect flower color variation among populations via local adaptation.

A second explanation is that the preference of certain pollinator groups could be dependent on the frequency of each morph in the population (i.e., the preferences for orange plants in Ancares are diluted in a context of yellow plants in Pontón). Empirical studies that examine plant species polymorphisms in flower color, morphology, or sex

**(A) Group 1 (*Bombus terrestris*)**

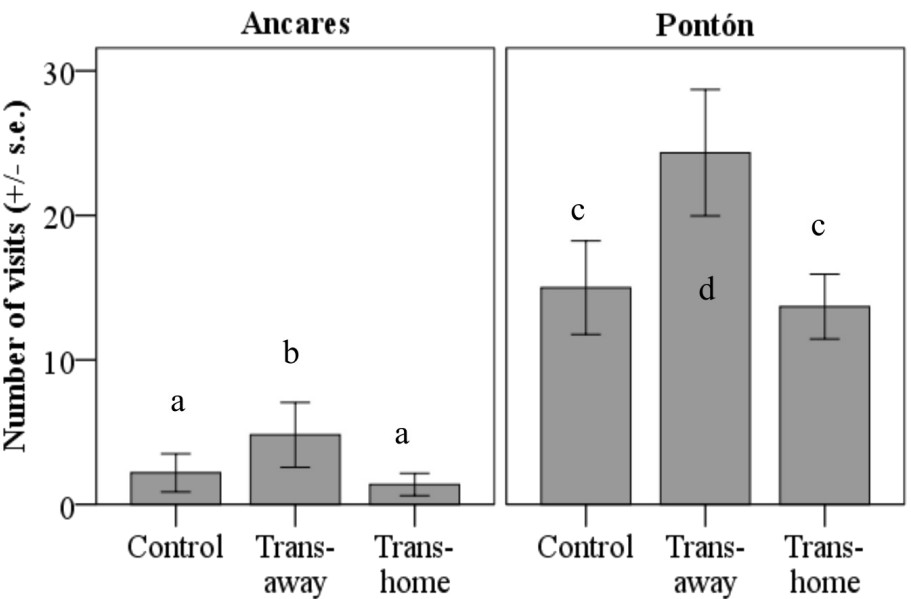

**(B) Group 3 (*Bombus pratorum, B. ancaricus and B. lapidarius*)**

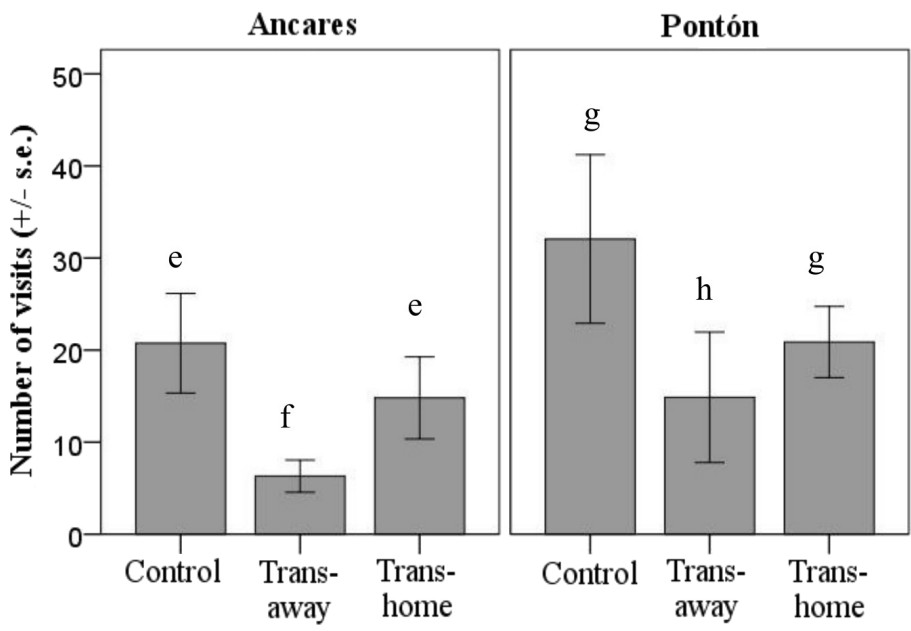

**Figure 4** **Number of floral visits per plant per 30 min (mean ± standard error) by two bumblebee groups (A) group 1, *Bombus terrestris*; (B) group 3, *B. pratorum, B. soroensis ancaricus, B. lapidarius decipiens*; depending on the locality at Northern Spain (Ancares, the orange morph population; and Pontón, the yellow morph population) and the experimental treatment (control, local transplant, foreign transplant).** Abbreviations: Trans-foreign, foreign transplant; Trans-local, local transplant. Different letters indicate statistical differences among treatments ($P < 0.01$).

phenotype generally show negative frequency dependence (see references in *Harder & Johnson, 2009*). However, in some cases this selection occurred somewhat heterogeneously and studies have found that common (or highly efficient) pollinators tend to exhibit positive frequency-dependent foraging, while less common (or less efficient) species might exhibit negative frequency-dependence (*Eckhart et al., 2006*) or positive frequency-dependence (*Malerba & Nattero, 2012*).

Our results show that differences in visitation rate between Ancares and Pontón populations depend on floral visitors from groups 1 and 3. Group 1 (*B. terrestris*) shows preferences for the least abundant morph in the population (orange in Pontón; negative frecuency-dependence). Conversely, group 3 shows a preference for the most abundant morph (orange in Ancares; positive frequency-dependence). Consequently, the preference of certain pollinator groups is dependent on the frequency of each morph in a population (*Smithson & Macnair, 1996*). Theory suggests that competition for floral resources might favor frequency-dependent foraging by some pollinator species, possibly contributing to the maintenance of flower color variation by frequency-dependent selection (*Gigord, Macnair & Smithson, 2001*; *Eckhart et al., 2006*).

Additionally, data from this work and previous studies show that the different pollinator abundance, especially groups 1 and 3, varies between years in Ancares and Pontón populations, but not selective pressures exerted on *G. lutea* flower color (see *Sobral et al., 2015*). However, the pollinator assemblage composition can also vary substantially year-to-year (e.g., *Herrera, 1988*; *Price et al., 2005*). Bumblebee foraging may differ among years (*Teräs, 1985*), among sites (*Elam & Linhart, 1988*; *Jones & Reithel, 2001*) and between queens and workers (*Teräs, 1985*; *Wesselingh & Arnold, 2000*). The floral color choices of bumblebees appear not to be governed by innate preference only, but also by environment conditions and colors of co-flowering plants (*Teräs, 1985*). Consequently, preferences for different color morphs may change temporally according to the community composition, generating a mosaic of selective pressures on floral color. Therefore, further analyses considering data from multiple years are needed to clearly support the hypothesis of flower color adaptation to local pollinator assemblages, even though possible technical and logistical challenges that would lead to the long-term monitoring of the transplant experiment.

In conclusion, the present study suggests no clear evidences of local adaptation to the pollinator community in each of the two *G. lutea* populations studied, even though some floral visitor groups (such as *Bombus pratorum*, *B. soroensis ancaricus* and *B. lapidarius decipiens*) consistently preferred the local morph to the foreign morph whereas others (such as *Bombus terrestris*) consistently preferred the foreign morph. Variation in pollinators foraging preference and visitation rate could generate a mosaic of frequency-dependent selection in *G. lutea* along the Cantabrian Mountains range. The consequences for local adaptation on *G. lutea* flower color would thus depend on variation in morph frequency, pollinator community composition and their effects on plant fitness.

## ACKNOWLEDGEMENTS

We are grateful to Paula Domínguez for the help in field work, Emilie Ploquin and José Ramón Obeso (University of Oviedo, Spain) for determining the bumblebee species, Brais Losada for the polishing edition of this manuscript. Luis Guitián drew the map of Fig. 2. We would like to thank the two anonymous reviewers for their suggestions and comments.

### Funding

This study is included in the project "Color polymorphism, geographic variation in the interactions and phenotypic selection. The case of *Gentiana lutea* L. in the Cantabrian Mountains," which was financially supported by Secretary of State of I+D+I, Ministry of Science and Innovation, Government of Spain (2009–2011). The funders had no role in study design, data collection and analysis, decision to publish, or preparation of the manuscript.

### Grant Disclosures

The following grant information was disclosed by the authors:
Secretary of State of I+D+I, Ministry of Science and Innovation, Government of Spain (2009–2011).

### Competing Interests

The authors declare there are no competing interests.

### Author Contributions

- Javier A. Guitián conceived and designed the experiments, performed the experiments, contributed reagents/materials/analysis tools, wrote the paper, prepared figures and/or tables, reviewed drafts of the paper.
- Mar Sobral conceived and designed the experiments, performed the experiments, analyzed the data, contributed reagents/materials/analysis tools, wrote the paper, prepared figures and/or tables, reviewed drafts of the paper.
- Tania Veiga conceived and designed the experiments, performed the experiments, reviewed drafts of the paper.
- María Losada wrote the paper, prepared figures and/or tables, reviewed drafts of the paper.
- Pablo Guitián and José M. Guitián performed the experiments, contributed reagents/materials/analysis tools, reviewed drafts of the paper.

### Field Study Permissions

The following information was supplied relating to field study approvals (i.e., approving body and any reference numbers):

Environmental Territorial Service from León, Regional Government of Castilla and León, Territorial Delegation of Government of Spain—Identifier:

12_LE_325_RNA_Puebla de Lillo_INV—Reference: 06.01.013.016/ROT/abp—File number: AEN/LE/103/12.

## Data Availability

The raw data has been supplied as a Supplementary File.

## Supplemental Information

Supplemental information for this article can be found online at http://dx.doi.org/10.7717/peerj.2882#supplemental-information.

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
