# Peer review of "Differences in pollination success between local and foreign flower color phenotypes: a translocation experiment with Gentiana lutea (Gentianaceae)"

_PeerJ, doi:10.7717/peerj.2882_

## Round 0.1 · original submission · Major Revisions

Thanks for submitting this MS. The subject matter looks quite interesting, and I would like to send it out for review as soon as possible. However, upon a quick reading of the MS, I note that there are quite a few language issues in the text. Prior to me finding referees and having them review the paper (and likely come to the same conclusion that I have regarding the language), could you please find an adept native speaker to give this MS a polishing edit?

Thank you, and I look forward to receiving the revised MS.

---

## Round 0.2 · Minor Revisions

The authors should consider both reviews carefully as both provide excellent advice for further improving this paper. Both reviewers were fine with the experimental approach and the treatment of the data. There were some concerns related to interpretation, and also a fair amount of editorial work on language by both. Neither reviewer suggests that more data are needed at this point or that the conclusions drawn are incorrect, rather both ask for some revision/refinement of the overall reporting and interpretation of the results in the larger context of pollination ecology, etc. While these things are always a judgement call, I believe that in this case the extent of the required revisions fit a bit better into the "minor" rather than the "major" category as none of the suggested changes require further experimentation or any substantial shift in analysis techniques or re-analysis.

Please work on these minor revisions and return a revised MS along with a detailed (line-by-line) rebuttal to PeerJ.

Note the attached doc from Reviewer 2.

Reviewer 1 ·

Basic reporting

1. The English could be improved, as I have indicated throughout my line comments.

2. Throughout the Materials and Methods section, I recommend that you change the passive voice to active voice. For example, at l. 92: “We measured the flower color of...”

3. In the Introduction, the authors could more clearly describe: (1) their previous work that suggests a relationship between variation in the pollinator community and flower color across G. lutea populations, and (2) their precise hypotheses or predictions for the study.

4. I suggest that you replace figure 3 with one that has slightly different data (but better addresses their question of how overall visitation rate varies between local and foreign flower color morphs). As it stands, figure 3 and 4 have some of the same data, and the question of how overall visitation to the local and foreign flowers is not presented clearly.

Experimental design

No comments.

Validity of the findings

The sampling is somewhat modest (reciprocal transplant between two populations, in only one year, 30 minutes per plant). Previous work in other plant-pollinator systems shows that visitation rates and the pollinator community can vary substantially from year-to-year, so I am concerned that the results would vary across years. The authors need to more clearly acknowledge and discuss this caveat, especially considering that between-population differences in the pollinator community is a central question of the work.

Additional comments

In this manuscript, the authors continue their work on understanding why flower color varies from yellow to orange in Gentiana lutea. It builds on work showing that flower color and the visiting pollinator community co-vary across plant populations (among other findings). In this study, the authors examine whether flower colors have adapted to their local pollinator communities, by carrying out a reciprocal transplant experiment between yellow and orange-flowered populations. They compared local and foreign flowers in terms of overall visitation rates and visits by different pollinator groups (mostly bumble bees, defined by tongue-length). There was no clear local morph advantage in overall visitation rate: local orange flowers received more visits than foreign yellow flowers in the orange population, while both local and foreign flowers received the same visits in the yellow population. However, particular pollinator groups preferred the local flower color morphs, while others preferred the foreign morphs. They suggest that these apparent differences in color-preference by different pollinators may contribute to the among-population variation in flower color.

The study is interesting and complements their previous work nicely. The work is interesting to me because the authors show how closely related pollinator species (within the same genus) may have different flower color preferences (of flowers that are otherwise the same in morphology) and that this may in part contribute to the flower color variation.

I think there also needs to be a more critical discussion of how variation in the pollinator community of such closely related organisms (bees in the same genus), could exert different selection pressures on flower color across populations. It is easy for me to understand how bumble bees with different tongue-lengths could exert different selection pressures on a trait like corolla length. However, flower color seems a bit trickier, as color vision seems like it should be fairly conserved at the generic level (I do not know that literature well enough, though, but I suggest you look into it). I think that flower colors in the surrounding plant community could play an important role (as Bombus tend to forage in a frequency-dependent way). Do flower colors in the plant communities vary across populations? (This is just a hypothesis.)

I have attached detailed corrections and comments on the manuscript. I hope they are helpful.

Annotated reviews are not available for download in order to protect the identity of reviewers who chose to remain anonymous.

·

Basic reporting

No issues on basic reporting.

Experimental design

Excellent experimental design.
One small suggestion to do a simple analysis on the number of floral visits received per plant (lines 170-173) for the different morphs at the two sites. For further info, see attached review document.

Validity of the findings

no comments

Additional comments

well written, elegantly designed study. A pleasure to review.

---

## Round 0.3 · accepted · Accept

Thank you to the reviewers for their comprehensive and helpful reviews that have improved this MS. And thanks to the authors who have done a good job of providing a detailed delineation of their revisions to the MS, both in the rebuttal and in the tracked-change document. I would encourage the authors to consider publishing the review history alongside this article, as PeerJ permits, if they feel comfortable doing so.